# Altered Host microRNAomics in HIV Infections: Therapeutic Potentials and Limitations

**DOI:** 10.3390/ijms25168809

**Published:** 2024-08-13

**Authors:** Maria J. Santiago, Srinivasan Chinnapaiyan, Kingshuk Panda, Md. Sohanur Rahman, Suvankar Ghorai, Irfan Rahman, Stephen M. Black, Yuan Liu, Hoshang J. Unwalla

**Affiliations:** 1Department of Chemistry and Biochemistry, Biochemistry Ph.D. Program, Florida International University, 11200 SW 8th Street, Miami, FL 33199, USA; msant206@fiu.edu (M.J.S.); yualiu@fiu.edu (Y.L.); 2Department of Cellular and Molecular Medicine, Herbert Wertheim College of Medicine, Florida International University, 11200 SW 8th Street, Miami, FL 33199, USA; schinnap@fiu.edu (S.C.); kpand014@fiu.edu (K.P.); mdsrahma@fiu.edu (M.S.R.); sghorai@fiu.edu (S.G.); stblack@fiu.edu (S.M.B.); 3Department of Environmental Medicine, University of Rochester School of Medicine and Dentistry, 601 Elmwood Ave., Rochester, NY 14642, USA; irfan_rahman@urmc.rochester.edu; 4Center for Translational Science, Florida International University, 11350 SW Village Parkway, Port St. Lucie, FL 34987, USA; 5Department of Chemistry and Biochemistry, Florida International University, 11200 SW 8th Street, Miami, FL 33199, USA

**Keywords:** microRNAs, HIV, host-virus interaction, therapeutics treatments, microRNA delivery systems

## Abstract

microRNAs have emerged as essential regulators of health and disease, attracting significant attention from researchers across diverse disciplines. Following their identification as noncoding oligonucleotides intricately involved in post-transcriptional regulation of protein expression, extensive efforts were devoted to elucidating and validating their roles in fundamental metabolic pathways and multiple pathologies. Viral infections are significant modifiers of the host microRNAome. Specifically, the Human Immunodeficiency Virus (HIV), which affects approximately 39 million people worldwide and has no definitive cure, was reported to induce significant changes in host cell miRNA profiles. Identifying and understanding the effects of the aberrant microRNAome holds potential for early detection and therapeutic designs. This review presents a comprehensive overview of the impact of HIV on host microRNAome. We aim to review the cause-and-effect relationship between the HIV-induced aberrant microRNAome that underscores miRNA’s therapeutic potential and acknowledge its limitations.

## 1. Introduction

Human immunodeficiency virus (HIV) is a high concern for public health worldwide. In the year 2023, the World Health Organization (WHO) reported that approximately 39 million people are currently infected with HIV. During 2023, approximately 600,000 people died from HIV-related infection, and 1.3 million people were killed from acquiring HIV, representing a significant threat to the worldwide population [1].

## 2. HIV and Its Genetic Variability

HIV is a member of the Retroviridae family, specifically within the lentivirus genus. Its genome is composed of two identical single strands of RNA [2]. Its genetic composition comprises nine overlapping genes expressed by alternative splicing: gag, polymerase (pol), and envelope (env), which serve as essential genes code for polyproteins (shared among all retroviruses), while tat and rev function as the regulatory proteins. Additionally, nef, vif, vpr, and vpu are the accessory genes. Two Long Terminal Repeats (LTR) flank these nine coding genes. Further translation of these nine coding genes will result in the production of the HIV 15 distinct proteins. The production of these 15 proteins is facilitated by the polyproteins that share overlapping open reading frames (ORFs), viral protease activities, and alternative splicing [3,4].

Despite only nine overlapping genes and two-flanked LTR, the HIV genome exhibits a high organization level and balance. This remarkable level of equilibrium mitigates the immunological selective pressure and the inherent fragility of the virus, particularly concerning mutations in its proteins [5]. This organization allows the virus to hijack or escape the immune system and rapidly develop drug resistance. Surprisingly, Dr. Cuevas’s group reported that a significant mechanism underlying genetic variability in HIV is not only the low fidelity of reverse transcriptase but also the host intrinsic cytidine deaminases of the A3 family. These deaminases account for a significant portion of the mutations observed in HIV, but the lower mutation frequency of around 2% plays a crucial role in HIV evolution [6]. While cytidine deaminases are categorized as antiretroviral factors because they hypermutate single-stranded viral DNA and inhibit viral replication, their activity also contributes to high genetic variability in the virus genome. This variability enables HIV to evade both therapeutic interventions and the host immune system [5,6,7].

## 3. HIV Life Cycle

The HIV life cycle is mainly divided into seven steps. The first step is binding the virus to the cell surface of target host cells, primarily CD4+ T lymphocytes (CD4+ T) and macrophages. HIV target cells express CD4+, the primary receptor for HIV, along with one or both co-receptors CCR5 or CXCR4, which form a trimolecular complex with the viral surface glycoprotein gp120 [8]. The second step is the fusion of the virus with the cell membrane, which takes only several minutes, and then the virus’s genomic RNA is released into cells protected by the viral capsid [8]. Once inside the cell, early replication begins with the reverse transcription of the virus’s RNA into DNA within the viral capsid. The capsid uses host cellular structures like microtubules and motor proteins (kinesin and dynein) to move toward and attach to the nuclear pore [9]. The capsid enters the nucleus after attaching to the nuclear pore, and the viral RNA is fully converted into double-stranded DNA. The capsid then breaks apart, releasing the finished DNA for integration into the host cell’s genome near the disassembly site, completing the third step [9,10].

The fourth step involves the integration of viral DNA into the host cell’s genomic DNA. HIV’s DNA integrates into the genomic DNA of the CD4+ T cell with the help of another of its viral protein integrases. Once integrated, the viral DNA is known as a provirus [8,11]. Following integration, the proviral DNA can be transcribed and translated, the fifth step, is to make the progeny virus, or the virus can remain dormant, establishing viral reservoirs [12,13]. Assembly and budding constitute the sixth and seventh (final) steps of the HIV life cycle. They involve the translocation of newly synthesized HIV genomic RNA and proteins to the cell surface, where they are assembled into nascent viral particles. The viral particles undergo the budding process from the host cell’s membrane. Finally, these particles detach, carrying a portion of the host cell membrane to initiate the next viral cycle [14,15].

HIV progresses to Acquired Immunodeficiency Syndrome (AIDS) when the CD4+ T-cell count decreases substantially (below 200 cells/mm^3^), leading to the inability of the body to fight off infections and diseases [16]. Figure 1 shows a schematic illustration of the HIV life cycle.

Above is a diagram illustrating the HIV life cycle initiation, starting with step 1 (binding). HIV binds to the surface of a CD4+ T cell utilizing the CD4+ receptors in conjunction with the co-receptor, CCR5, forming the trimolecular complex with the viral surface glycoprotein gp120. In step 2, (fusion) the viral membrane fuses with the host cell membrane, allowing the virus to enter the cell. Step 3 (reverse transcription and nuclear entry) demonstrates that the virus employs a reverse transcriptase enzyme to start the conversion of its RNA genome into the DNA within the host cell’s cytoplasm protected by its viral capsid. Simultaneously, the viral capsid recruits host microtubules and motor proteins to facilitate nuclear entry. In step 4 (integration), using its enzyme integrase, the viral DNA becomes permanently incorporated into the host cell’s genome, becoming an integral part of its genetic material. In step 5 (transcription and translation), the host cell machinery transcribes and translates the integrated viral DNA, generating new viral RNA and proteins, thereby facilitating the replication process. In step 6 (assembly), new viral RNA and proteins are assembled into complete viral particles within the host cell. Finally, in step 7 (budding), the newly assembled viral particles acquire lipid envelopes from the host cell membrane. They are released into the extracellular space, ready to infect other cells and continue infection.

## 4. cART and Its Limitations

The introduction of a combination of antiretroviral therapy (cART) in 1996 marked a pivotal moment for controlling the AIDS epidemic by enabling the effective management of HIV infection [17]. To date, cART remains the most effective HIV treatment, primarily due to the synergistic effects of its multiple drug components designed to target various stages of the HIV life cycle [17,18]. cART has notably improved the quality of life and increased the lifespan for people living with HIV (PLWH) [19,20,21]. However, cART has its limitations; while it is effective in controlling HIV to the point that HIV is undetectable, it is not a cure. Hence, a life-long commitment to cART is required to maintain viral suppression. This raises the concern that prolonged exposure to cART may result in toxicity and the emergence of resistance over time [18]. cART can also lead to adverse drug interactions when used in combination with other drugs used to treat common comorbidities of PLWH [22]. In addition, the cost of cART is extremely expensive [18,23]. A meta-analysis conducted in the United States by Dr. McCann and colleagues in 2020 revealed an average cost of approximately USD $36,000 per patient per year [24]. These findings underscore the critical need for ongoing research into innovative therapeutic interventions to develop new alternative treatments for HIV that offer improved efficacy and lower costs.

## 5. HIV-Causing Cellular Metabolic Dysregulations via microRNAs

Understanding the crosstalk between HIV and cellular factors is critical to the development of therapeutic approaches against HIV/AIDS. It is known that HIV induces significant metabolic alterations in the host genome. These alterations include shifts in gut microbiota composition, which affect nutritional dynamics, and modifications in glycerophospholipid (GPL) metabolism and metabolites. Such alterations disrupt bioenergetics, elevate oxidative stress, and dysregulate ion homeostasis and ferroptosis, exacerbating immune system responses and mitochondrial dysfunction [25,26,27,28,29]. Extensive research has highlighted the significance of microRNAs in HIV pathogenesis [30,31,32,33,34,35,36,37]. Certain microRNAs can either enhance viral infection or impede viral replication by targeting the HIV-pro-viral genome or regulating host genes crucial for viral replication and immune responses, summarized in Table 1.

Furthermore, microRNAs are endogenous noncoding molecules; thus, our biological system may have less chance of developing drug resistance to microRNA [53,54]. MicroRNAs’ primary function is to suppress gene expression, as they serve as the guiding sequence for the RNA-Induced Silencing Complex (RISC) [55]. For these reasons, they have immense potential to be applied as therapeutics.

## 6. Mechanisms Underlying HIV-Induced Aberrant microRNAome

The mechanisms by which HIV alters the host’s microRNAome remain not fully elucidated yet. One particular mechanism includes actions of Tat and Vpr viral proteins, which can activate CCAAT/enhancer-binding protein beta (C/EBP β). This DNA-binding protein can modify the expression of microRNAs [39]. Also, Tat is an RNA binding protein responsible for recruiting the host positive transcription elongation factor b (P-TEFb) to the Trans-activation response (TAR) RNA hairpin formed at the 5′-end of the nascent viral RNA, P-TEFb phosphorylates the S2 residues of RNA Polymerase II (Pol II). Pol II is situated at the 5′-end RNA transcripts to activate the transcription of microRNA genes [56,57,58]. HIV-1 Tat can activate nuclear factor kappa beta (NF-κβ) by directly interacting with SIRT1 and blocking SIRT1’s ability to deacetylate lysine 310 in the p65 subunit of NF-κβ, leading to its activation [59]. NF-κβ is a master transcription factor that modulates the expression of multiple microRNAs [49,59]. HIV Tat upregulates the transforming growth factor beta-1 (TGF-β1) signaling, which activates the canonical SMAD-2 and SMAD-3, and the non-canonical signals, which include TGF-β/mitogen-activated protein kinases (MAPK), TGF-β/SMAD-1/5, and TGF-β/phosphatidylinositol-3-kinase/Akt that in turn to modulate microRNA expression [43,60,61,62,63]. A summary of these pathways is illustrated in Figure 2.

HIV proteins trigger signal transduction pathways, leading to modulation of specific miRNAs. This dysregulation primarily manifests as direct downregulation of target genes via RISC, facilitated by upregulated miRNA expression, or as the upregulation of genes due to the downregulation of their targeted miRNAs.

## 7. Alteration of microRNA Expression Profiles by HIV

The Tatro group has utilized a microRNA array and conducted in vitro experiments to demonstrate substantial alterations in microRNA profiles among PLWH of patients with both HIV and major depressive disorder (M.D.D.) when compared to non-infected controls. Interestingly, the microRNA profiles of HIV/M.D.D. patients displayed significant similarities with those seen in Alzheimer’s disease. Moreover, miR-125a and miR-22 exhibited notable upregulation in PLWH. MiR-125a targets the intracellular membrane protein IFITM3, while miR-22 targets sTNFR1A, a secretory protein implicated in neuroinflammation. The upregulation of these microRNAs leads to significant downregulation of their cognate target proteins that are critical in developing M.D.D. [38]. This highlights the potential use of microRNAs as possible biomarkers.

Sun and colleagues used microRNA arrays to study how the microRNAome profile of the host changes upon HIV infection. The data have shown that miR-223 levels were significantly upregulated in HIV-1-infected CD4+CD8−peripheral blood mononuclear cells (PBMCs), whereas miR-29a/b, miR-155, and miR-21 levels were significantly downregulated. They further demonstrated that the HIV-1 Nef-3′-LTR region serves as the 3′-UTR for all HIV-1 transcripts, suggesting that miRNAs sharing the identical seed sequences in this region play a critical role in regulating the expression of HIV proteins [39]. In addition, several studies have shown that the miR-29 microRNA family plays a vital role in suppressing viral load [40,41].

Yuan and colleagues demonstrated that treating monocyte-derived macrophages with HIV proteins such as Tat and gp120 altered the expression of microRNAs miR-27a and miR-21, leading to a significant disruption of intracellular tight junctions and dysfunction of mitochondrial biogenesis. These events, in turn, stimulate glycolysis and promote an inflammatory response [42].

Likewise, Reynoso and colleagues showed that chronic HIV infection in PLWH is associated with downregulation of microRNAs miR-29b-3p, miR-33a-5p, and miR-146a-5p compared to uninfected control and elite suppressors (HIV patients with a robust immune system and without symptoms). The measurement of p24, an index of viral load, demonstrated that these microRNAs significantly impact virus replication. In addition, Reynoso’s group also showed that these microRNAs mainly target viral DNA integration by downregulating Nef, JNK, and CXCR4 proteins. The suppression of the above microRNAs could contribute to the enhanced pathogenicity of HIV [36].

Similarly, the studies from our group have also shown that HIV Tat and TGF-β1 downregulate miR-141-5p in bronchial epithelium in vitro. The suppression of miR-141-5p can increase the expression of its cognate target, CCR5, thereby enhancing viral entry into cells [43].

In addition, Qiao and colleagues demonstrated that HIV-1 infection can trigger the upregulation of miR-210-5p, suppressing TGIF2 and consequently causing cells to fall into G2 cell cycle arrest. They also showed that the HIV Vpr protein facilitates the upregulation of miR-210-5p. This upregulation is advantageous for HIV because the induction of the G2 cell cycle arrest in host cells allows HIV to transcribe its genes at an accelerated rate, thereby heightening its pathogenicity [44,64].

HIV significantly impairs the immune system, which further facilitates immune evasion. Fan and colleagues have reported that miR-144 expression can be increased in alveolar macrophages (A.M.s) from HIV-1 transgenic rats and in HIV-1-infected human monocyte-derived macrophages (M.D.M.s) compared to cells from W.T. rats and non-infected human M.D.M.s, respectively. Increasing miR-144 can suppress Nrf2, impairing bacterial phagocytic activity and H_2_O_2_ scavenging ability [45].

Ratifying its inflammatory effect, HIV increases the expression of miR-88 and miR-99, promoting the activation or release of pro-inflammatory molecules like tumor necrosis factor-alpha (TNFα), interleukin (IL)-6, and IL-12 and TLR8. This cytokine storm dysregulates the immunological response [46,65]. Similarly, our group has also shown that miR-145-5p and miR-509-3p can be upregulated in primary bronchial epithelial cells upon HIV infection, suppressing CFTR and disrupting the mucociliary clearance (MCC) in the lungs [35].

HIV can promote and activate its replication by altering the microRNAome. HIV downregulates miR-196b, miR-1290, and miR-149, which have identical seed sequences and binding sites for the HIV-1 3′ UTR, thereby modulating the expression of viral proteins such as Vpr and enhancing HIV pathogenicity [47,48]. On the other hand, HIV upregulates the expression of miR-34 and miR-182, suppressing the SIRT-1 and NAMPT, leading to increased acetylation of the p65 subunit of NF-κβ. Consequently, this promotes the transcription of HIV genes from the LTR [49,50]. In a severe scenario, HIV downregulates the expression of miR-150-5p, miR-198, and miR-27b, leading to upregulation of SOCS 1 and Cyclin T-1 gene expression and robust HIV replication. Cyclin T-1 is a positive modulator of HIV transcription elongation, and SOCS1 stimulates inflammatory mediators to trigger RNA binding protein (TREM1/CIRP) receptors, which are pro-inflammatory cytokines amplifiers. These events exacerbate the immune response, increasing immune cell recruitment and enhancing HIV replication [51,52,66]. A summary of all the microRNAs and their function in regulating gene expression discussed is illustrated in Table 1 and Figure 2.

## 8. Considerations for Interpreting microRNA Alterations in HIV Studies

It is crucial to distinguish the results generated from in vivo and in vitro experiments when analyzing results. In vitro studies often involve a high proportion of cells infected with HIV, whereas patient samples in vivo typically have only a small fraction of infected cells. This discrepancy can significantly impact the observed effects and their interpretations [67,68]. For example, the in vitro study by Yuan and colleagues showed that treatment of monocyte-derived macrophages with HIV proteins such as Tat and gp120 can result in altered expression of miR-27a and miR-21, leading to significant cellular disruptions such as mitochondrial dysfunction and increased glycolysis [42]. However, this study uses a controlled environment in laboratory settings where a high proportion of cells are intentionally infected or treated with HIV proteins. Thus, the experimental conditions may not accurately reflect normal biological conditions. Similarly, Qiao and colleagues have found that HIV-1 infection upregulated miR-210-5p in cultured cells, causing G2 cell cycle arrest and enhanced viral transcription [44]. The results from the in vitro studies may not be fully translated to the clinical context, where only a small proportion of cells are infected. The in vitro and in vivo discrepancy may significantly influence the observed effects and the interpretations of results. Understanding the difference is essential as it may affect the generalizability and interpretation of the findings. Regarding in vivo models, several factors need to be considered. These include a low proportion of infected cells compared with in vitro studies and patient heterogeneity, including variations in disease progression, treatment status, and co-infections. Reynoso and colleagues have investigated the effects of chronic HIV infection on microRNA levels in patient samples. They have shown that microRNAs, miR-29b-3p, miR-33a-5p, and miR-146a-5p—were downregulated. These microRNAs play important roles in regulating various cellular processes, including viral replication and the integration of viral DNA into the host genome [36]. However, it should be noted that the study uses patient samples, where typically only a small fraction of cells are infected with HIV. Thus, while the study provides valuable insights into how these microRNAs are affected by HIV in a clinical setting, the findings may not fully reflect the dynamics observed in laboratory conditions where a higher proportion of cells can be infected. In other words, the low proportion of infected cells in the patient samples may influence the extent to which the microRNAs and their target proteins impact viral replication and integration. This limitation underscores the importance of considering how in vivo conditions, such as patient heterogeneity and the low infection rate, may affect the interpretation of results and the applicability of findings to broader contexts. Similarly, Tatro’s group reported significant alterations in microRNA profiles among PLWH with both HIV and major depressive disorder, with similarities to Alzheimer’s disease profiles [38]. This also suggests the interplay between HIV infection, co-morbid conditions, and microRNA regulation in patients for the interpretation of results in in vivo models.

It should also be noted that it is crucial to consider that in vivo studies often recruit patients receiving a combination of antiretroviral therapy (cART), which can impact RNA metabolism and microRNA profiles [69,70]. For instance, Reynoso and colleagues have analyzed plasma samples from 27 subjects who were divided into three groups: 10 Elite Controllers (individuals with a plasma viral load < 50 copies/mL and CD4+ count > 350/mL, not on cART), 10 chronic HIV patients on cART, and seven healthy donors. They have found that the plasma miRNA pro-file can distinguish between Elite Controllers and chronic HIV patients on cART. Specifically, they have observed that levels of hsa-miR-29b-3p, hsa-miR-33a-5p, and hsa-miR-146a-5p are higher in elite controllers’ plasma than in chronic HIV patients [36]. This suggests that cART can influence microRNA expression, potentially affecting viral replication and integration. This further suggests that the treatment status of the patients in this study can influence microRNA levels and their interactions with target proteins, highlighting the need to consider cART when interpreting results.

## 9. Limitations of microRNA-Mediated Disease Therapy

Evidence shows that HIV mainly alters host microRNAs to facilitate its transcription and replication (see Table 1). Multiple studies demonstrate that microRNA mimics or inhibitors can mitigate the adverse effects of HIV [35,36,37,39,40,42,43,44,47,48,49,50,52,71]. This underscores the use of microRNAs as candidates for disease therapy development. However, there are several limitations to the use of microRNAs in disease therapy. Figure 3 shows multiple challenges in the development of microRNA-based disease therapy.

MicroRNA-based therapies face challenges maintaining stability within biological environments, impacting their efficacy over time: Immune response: the immune system’s recognition of modified microRNAs or carrier molecules can trigger immune responses, potentially hindering therapeutic outcomes; delivery system design and cost: designing efficient delivery systems for microRNA therapeutics involves addressing cost constraints alongside ensuring effective targeting and uptake; delivery systems: carrier molecules facilitating microRNA delivery exhibit diverse natures, including both viral and non-viral systems, each with advantages, disadvantages, and specific characteristics; off-target gene effects: non-specific interactions of microRNAs may lead to unintended gene silencing; body elimination and circulation: microRNAs encounter obstacles to circulation and elimination processes within the body, affecting their distribution and exposure time at target sites; and modification requirements: additional modifications may be necessary to enhance the stability, specificity, and delivery efficiency of microRNA-based therapies, adding complexity to their development process. 

### 9.1. Summary of Common Limitations to Developing microRNA-Based Therapy

microRNA-based therapies have several challenges that need to be addressed. First, maintaining stability in biological environments is challenging, impacting the long-term effectiveness of these treatments. The immune system can recognize modified microRNAs or carrier molecules, potentially triggering immune responses and hindering therapeutic outcomes. When designing delivery systems for microRNA therapeutics, it is essential to consider both cost constraints and the effectiveness of targeting and uptake. The carrier molecules used for delivery vary widely, including both viral and non-viral systems, each with pros, cons, and unique characteristics.

Additionally, off-target effects are a significant concern since non-specific interactions of microRNAs can lead to unintended gene silencing. microRNAs also face challenges in circulation and elimination within the body, affecting their distribution and duration at target sites. Further modifications may be necessary to enhance stability, specificity, and delivery efficiency, adding to the complexity of the development process.

### 9.2. Stability and Prompt Degradation of microRNA

Antagomirs and mimics of microRNAs are susceptible to degradation by ribonuclease enzymes within the serum or cells. Naked miRNAs, characterized by an unmodified 2′-OH in the ribose moiety, undergo rapid degradation, often within seconds, by nucleases such as serum RNase A-type nucleases in the blood. Thus, the challenge in developing therapeutics lies in achieving a stable molecule that can withstand delivery [71,72].

### 9.3. Immune Response and Off-Target Effects

Our innate immunity recognizes oligonucleotides as pathogen-associated molecular patterns (PAMPs). Toll-like receptors (T.L.R.s) are part of the pattern recognition receptors (P.R.R.s), which can sense double- and single-stranded oligonucleotides. This recognition can develop immunity against microRNA during microRNA-based therapy [54,73,74,75]. Another challenging issue is that in their inherent nature, microRNAs have multiple target genes and can simultaneously alter the expression of multiple genes. This characteristic of microRNAs allows them to inadvertently target non-target genes, leading to undesirable alterations in gene expression with deleterious effects [72,76].

### 9.4. Modifications and Delivery Systems of microRNAs

Oligonucleotides do not penetrate the phospholipid cell membranes due to their hydrophilic nature and low stability [54,77]. Therefore, the likelihood of administering naked microRNAs without a delivery system that provides stability is minimal. The most common delivery systems include modifying the microRNA or a carrier molecule to enhance stability and proper delivery.

### 9.5. Common Modifications of microRNAs

Some of the most common modifications of microRNAs are substitutions in the 2′ carbon of the ribose sugar by 2′-O-methyl (2′-OMe) or 2′-O-methoxyethyl (2′-MOE), which increase the binding affinity to proteins and nuclease resistance due to the conformational shift in the sugar moiety [78].

In addition, substituting non-bridging oxygen atoms in the phosphate backbone with sulfur can alter the backbone’s charge, enhancing the backbone’s stabilization. This modification is referred to as the phosphorothioate nucleotides (P.S.) [77,79]. Likewise, introducing a methylene bridge between the 2′-oxygen and 4′-carbon of the ribose moiety, commonly referred to as a Locked Nucleic Acid (L.N.A.) modification or adopting the “locked” conformation, enhances backbone rigidity. This modification reduces susceptibility to nuclease degradation and enhances base stacking stability, resulting in elevated melting temperatures (Tm). Additionally, it aids in preorganization for base pairing, facilitating efficient binding [77,78,80].

### 9.6. Delivery Systems of microRNAs: Carrier Molecules

#### 9.6.1. Lipid-Based Formulations: Nanoparticles or Liposomes

Lipid nanoparticles and liposomes offer encapsulation and protection to microRNAs within their lipid bilayer or aqueous core. This shielding protects miRNAs from nucleases, enhancing their stability. Their lipid nature facilitates cellular uptake by promoting lipid fusion with the cell membrane. Crucially, the delivery systems can be engineered to target specific cell types or tissues by incorporating targeting ligands or antibodies. This specificity enables selective binding to receptors or antigens expressed on target cells, thereby reducing the risk of off-target effects. However, they have limitations, such as limited payload capacity, toxicity, storage instability, cost-production, and batch variance [81,82].

#### 9.6.2. Polymer-Based Formulations

Polymers like polyethylene glycol (P.E.G.) or poly(lactic-co-glycolic acid) (P.L.G.A.) can be used to formulate microRNAs into nanoparticles, providing protection and controlled release properties and biocompatibility. Polymer-based nanoparticles control microRNAs’ release by modulating their polymer matrix’s properties, such as their composition or degradation rate. In addition, they possess a high versatility, allowing them to accommodate a wide range of cargo, such as nucleic acids, drugs, or fluorescence markers [83]. However, they are limited by polymers’ complex synthesis processes, their stability, and the spontaneous release of cargo in some cases [84].

#### 9.6.3. Viral Vectors

Viral vectors, such as adeno-associated viruses (AAVs) or lentiviruses, can be engineered to deliver miRNAs to target cells efficiently. However, the microRNA expression cassette encoding the desired miRNA sequence has to be inserted into the viral vector genome. Then, the viral vector has to be purified and administered to target cells in vitro or in vivo. The long process of creating the viral delivery system may cause genetic mutation, production challenges, and immune toxicity [85,86].

#### 9.6.4. Exosome-Based Delivery

MicroRNAs can be packaged into exosomes, natural extracellular vesicles, for delivery to target cells (functionalized exosome), providing a natural and potentially safer delivery mechanism [87]. However, their isolation and purification processes are complex. They also present a specific bio-distribution and targeting specificity with variability between their sizes and can invoke an immune response [88,89].

### 9.7. Body Elimination

Modifications on microRNA, such as the 2′-O-methoxyethyl (2′-MOE), confer stability and resistance to degradation of microRNAs, prolonging their half-life. This modification leads to an accumulation of the microRNAs, mainly in the liver and kidneys, altering their elimination from the body, and it could lead to the disruption of metabolic and excretory processes [90,91,92]. Figure 4 summarizes the microRNA delivery system’s major microRNA modifications and carrier molecules.

The microRNA modifications show the modified moiety in red. Carrier molecules are shown as general representations, with the microRNAs as cargo.

## 10. Conclusions

In summary, the role of microRNAs as possible biomarkers and treatment of HIV is promising. This review summarizes evidence that specific microRNAs are dysregulated in the host upon viral infections. These microRNAs hold the potential to serve as targeted therapies for various HIV comorbidities. They can act as precision tools or “magic bullets” to combat specific AIDS and non-AIDS comorbidities in PLWH. For instance, certain microRNAs can target and inhibit critical genes involved in HIV replication (see Table 1). Delivering synthetic versions of these microRNAs into cells may suppress viral replication and reduce the spread of the virus.

In addition, microRNAs can also regulate the host’s immune response to viral infection. By manipulating the expression of specific microRNAs, it may be possible to enhance the immune system’s ability to suppress HIV and reduce its pathogenicity. Overall, microRNA-based therapy for HIV is still in the early stages of development, but it holds significant potential for the treatment of HIV.

While there is not yet a direct miRNA-based therapy specific to HIV, there are ongoing clinical trials to treat different diseases with microRNA-based therapy. For instance, anti-miR-103 and anti-miR-107 (Clinical trial Identifier: NCT02612662, NCT02826525) are in phase I and II to treat type II diabetes [93]. Anti-miR-122 (Clinical trial Identifier NCT01200420) is in phase II to treat hepatitis virus C, which is a common co-infection in HIV [93,94,95]. In addition, there is the 2′-MOE modification approach of microRNAs (Clinical trial Identifier NCT02981602) in phase II to treat hepatitis B virus [93]. These provide evidence that there is a future for the development of microRNA-based therapy for HIV. However, additional research is necessary to gain a deeper understanding of the mechanisms and implications of these noncoding regulatory elements and optimize their pharmacokinetics and pharmacodynamics.

## 11. Future Directions

Future directions should focus on advancing microRNA-based treatments by ensuring the stability of microRNAs to prevent degradation by RNA nucleases, enhancing the specificity of target genes beyond just the seed sequence to mitigate the off-target effects of microRNAs, and precision targeting of the organ requiring treatment. Since each organ comprises various cell types, microRNAs may elicit different responses depending on the cell type present.

Due to their low stability, microRNAs require a carrier to reach specific tissues. While microRNAs are naturally present within cells and should not be toxic to tissues, the delivery system may exhibit toxicity. Further research and optimization are necessary to develop a successful microRNA delivery system.

In addition, clinical studies should take into account multiple factors, such as the nature of the experiment (in vivo or in vitro), the conditions and interactions with cART, and patient heterogeneity.

Moreover, alternative non-human microRNA-based approaches need to be developed. For example, specific microRNAs derived from plants, which can affect pathogen-related gene expression but do not target host genes, may be designed as new approaches for microRNA-based disease therapy.

## Figures and Tables

**Figure 1 ijms-25-08809-f001:**
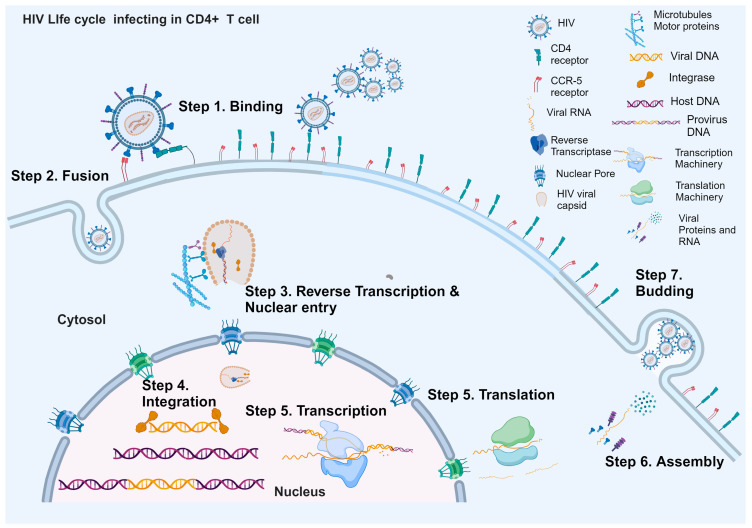
Schematic representation of HIV life cycle.

**Figure 2 ijms-25-08809-f002:**
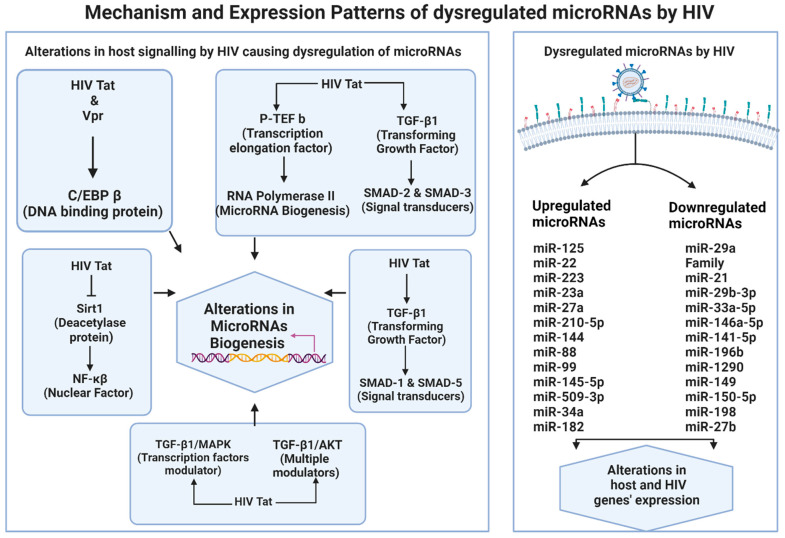
Mechanisms and expression patterns of dysregulated microRNAs by HIV.

**Figure 3 ijms-25-08809-f003:**
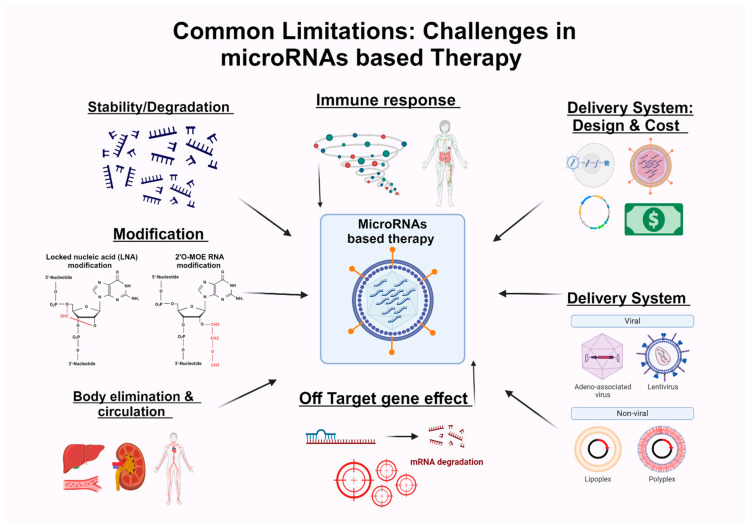
Common limitations to developing microRNA-based therapy.

**Figure 4 ijms-25-08809-f004:**
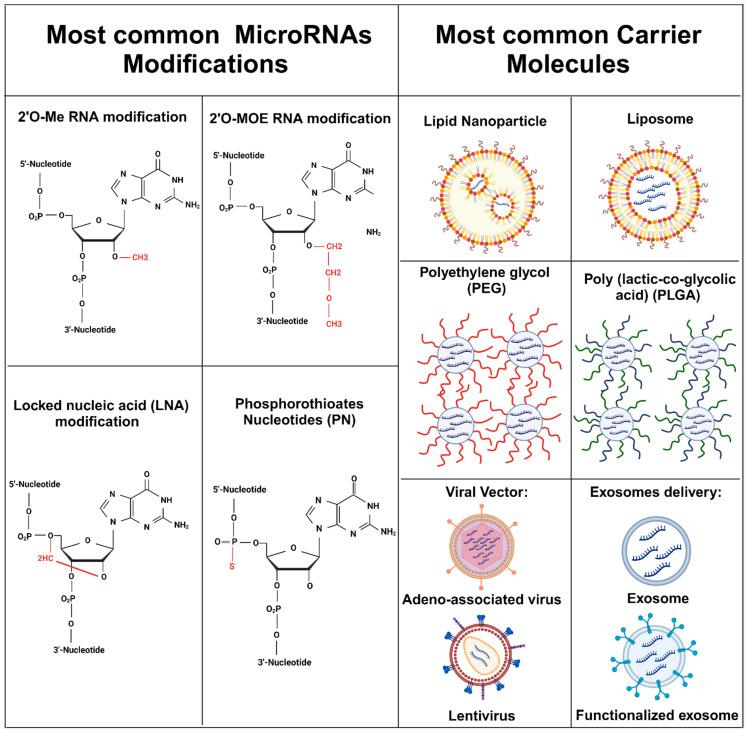
Most common microRNA modifications and their carrier molecules.

**Table 1 ijms-25-08809-t001:** Alteration of microRNA profiles by HIV infection.

MicroRNAs	Expression Level	Target	Alteration	Refs.
miR-125a miR-22	Up Up	Interferon Induced Transmembrane Protein 3 (IFITM3) Soluble Tumour Necrosis Factor Receptor I (sTNFR-1A)	Intracellular protein suppression Alteration of neuroinflammation	[38]
miR-223	Up	Viral Nef	Expression of HIV proteins Apoptosis	[39]
miR-29a family miR-155 and miR-21	Down	3′-untranslated region of HIV-1 mRNA	Degradation of viral mRNA	[39,40,41]
miR-23a miR-27a	Up	Zonula occludens (ZO-1) Peroxisome proliferator-activated receptor gamma (PPARγ) Glucose transporter 1 (Glut1)	Disruption of tight junction Dysfunction of mitochondrial biogenesis Increase in an inflammatory response	[42]
miR-29b-3p, miR-33a-5p miR-146a-5p	Down	Viral protein Nef MAPK8 CXCR4 receptor	Promote Viral replication Increasing pathogenicity, viral DNA integration, and entry	[36]
miR-141-5p	Down	C-C chemokine receptor type 5 (CCR5)	Increase viral Infection	[43]
miR-210-5p	Up	TGF-β-induced factor homeobox 2 (TGIF2)	G2 cell cycle arrest/Increases expression of viral genes	[44]
miR-144	Up	Nuclear Factor (erythroid-derived 2)-like 2 (Nrf2)	Increased oxidative stress decreases the immune response	[45]
miR-88 & miR-99	Up	Toll Like Receptor 8 (TLR8)	Increase the release of cytokines TNFα, IL-6, and IL-12	[46]
miR-145-5p and miR-509-3p	Up	Cystic Fibrosis Transmembrane Conductance Regulator (CFTR)	Suppressed MCC and recurrent lung infections and inflammation	[35]
miR-196b and miR-1290	Down	HIV-1 3′ UTR for viral expression	Promote HIV expression	[47]
miR-149	Down	Viral protein Vpr	Promote viral infection	[48]
miR-34a	Up	Sirtuin 1 (SIRT1)	Promotes Tat-induced HIV-1 transactivation	[49]
miR-182	Up	Nicotinamide phosphoribosyltransferase (NAMPT)	Promote Tat-induced LTR transactivation	[50]
miR-150-5p	Down	Suppressor of Cytokine Signaling (SOCS1)	Promote HIV-1 replication and reactivation	[51]
miR-198 and miR-27b	Down	Cyclin T-1	Robust HIV replication	[52]

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
