# Peer review of "Altered Host microRNAomics in HIV Infections: Therapeutic Potentials and Limitations"

_ijms, 2024, doi:10.3390/ijms25168809_

Round 1

Reviewer 1 Report

Comments and Suggestions for Authors

Santiago et al summarize the emerging role of microRNAs in HIV infection and discuss their viability as a potential therapeutic approach to complement existing ART. In general, the writing quality, and some content, needs to be improved throughout. Some of the information regarding the HIV lifecycle and the role of APOBEC3s in virus mutation are not quite accurate. Overall this is an interesting summation of the research focused on understanding the olre of microRNAs in HIV biology.

- In general, the writing quality needs to be improved throughout. Many of the statements are fragmented/incomplete thoughts, have inappropriate noun-pronoun agreements, or use incorrect scientific phrasing.  

- Section #1 should clarify that the numbers referenced for people that have died from HIV-related infections and AIDS would be from a single year, and not cumulative, which is how it currently reads.

- Polymerase does not serve a structural function as stated on lines 42-43. It comprises the viral enzymes reverse-transcriptase, protease, and integrase.

- Sentence 46 does not make sense as written. Gag and Pol polyproteins are liberated by protease cleavage, the individual proteins are not released by “further translation” as stated in the text.

- The statements made on lines 56-58 about mutation frequency and HIV are not entirely accurate. Yes, the referenced study determined that a high degree of mutation frequency arises from APOBEC3-mutation, but that was for hypermutated/dead virus, which highlights the antiviral restriction capacity of these enzymes. This study concluded that it is the 2% mutation frequency, not the 98% as the authors state, that contributes to HIV evolution. Also, the APOBEC3s that can restrict HIV replication predominantly hypermutate single-stranded viral DNA, not RNA as the author’s state. These statements should be corrected or removed.

- The mechanism of reverse-transcription and nuclear import of the viral capsid core has been updated in recent years. The authors should revise section #3 and figure #1 to reflect the current working model of the virus lifecycle (i.e, the naked viral RNA is not reverse-transcribed in the cytoplasm).

- Not sure if lines 115-122 are meant to be separate one-sentence paragraphs?

- It could be worth separating the observations in section #7 into direct and indirect effects of HIV infection. As it reads now, it’s unclear how each observation stratifies. For example, one would assume the changes to bronchial epithelial cells would be an indirect effect of infection as opposed to changes in MDMs or T-cells.

- It seems as though the paragraphs from lines 245-258 and 259-273 are duplicated?

Comments on the Quality of English Language

In general, the writing quality needs to be improved throughout. Many of the statements are fragmented/incomplete thoughts, have inappropriate noun-pronoun agreements, or use incorrect scientific phrasing.  

Author Response

Dear Reviewer 1,

Thanks!

Reviewer 2 Report

Comments and Suggestions for Authors

“Altered host microRNAomics in HIV infections: Therapeutic Potentials and Limitations”

The review is well written and provides an interesting overview of the microRNAomics in HIV infections. Some microRNAs can either favor HIV replication or inhibit viral replication by targeting HIV proviral genome or host genes crucial for viral replication and immune response. Therefore, microRNAs based therapies could complement the actual combined antiretroviral therapy to suppress HIV replication. However, we are still far from practical applications of microRNA-based therapies as illustrated and highlighted in this review.

Are the authors aware of any ongoing clinical trials involving the use of microRNAs in people living with HIV? This information, if available, could be summarized in a short paragraph.

Author Response

Dear Reviewer 2,

Thanks!
